# Facilitators and barriers to use rapid antigen test for SARS-CoV-2 among community-dwelling older adults in Hong Kong: A population-based cross-sectional random telephone survey

Joseph Kawuki[1], Yuan Fang[2], Fuk-yuen Yu[1], Danhua Ye[1], Paul Shing-fong Chan[1], Siyu Chen[1], Zixin Wang[1] *

1 Centre for Health Behaviours Research, Jockey Club School of Public Health and Primary Care, The Chinese University of Hong Kong, Hong Kong, China, 2 Department of Health and Physical Education, the Education University of Hong Kong, Hong Kong, China

* wangzx@cuhk.edu.hk

## Abstract

Faster diagnosis of Coronavirus disease 2019 (COVID-19) is crucial for surveillance, prompt implementation of infection control measures and adequate patient care among older adults. This study investigated the behavioral intention to use Rapid Antigen Tests (RAT) and associated factors among older adults in Hong Kong for health monitoring and when having COVID-19-like symptoms. This was a population-based cross-sectional random telephone survey of 370 Chinese-speaking adults aged ≥65 years. The behavioral intention to use RAT was the main outcome, and logistic regression models were used to assess the associated factors, using SPSS (version 26.0.). Results indicate that among the participants, 90.3% had used RAT, of which 21.6% obtained positive results. The common challenges faced when using RAT included: difficulty choosing the right RAT kit, uncertainty about how to use RAT, and not knowing what to do after getting a positive result. Additionally, 27.3% intended to use RAT regularly for health status monitoring without any symptoms, while 87.0% if they had COVID-19-like symptoms. After adjustment for significant background characteristics, positive attitudes, perceiving Hong Kong government and their children and/or other family members would support them using RAT, belief that RAT health promotion materials were helpful to understand how to use RAT and thoughtful consideration of the veracity of COVID-19 specific information were associated with higher behavioral intention to use RAT both when having no symptoms and in presence of COVID-19-like symptoms. Having negative attitudes toward RAT was associated with a lower intention of RAT use only when having no symptoms. Addressing difficulties faced when using RAT, strengthening positive attitudes, involving significant others and empowering with adequate information-veracity evaluating skills are potentially vital strategies to increase RAT use among older adults.

**Data Availability Statement:** All relevant data underlying the findings described in the manuscript is within the paper.

**Funding:** This work was supported by Health and Medical Research Fund, Food and Health Bureau, Hong Kong Special Administrative Region (19181152 to ZW). The funders had no role in study design, data collection and analysis, decision to publish, or preparation of the manuscript.

**Competing interests:** The authors have declared that no competing interests exist.

# Background

Diagnostic testing is a critical component of the overall prevention and control of coronavirus disease 2019 (COVID-19). Laboratories have been using nucleic acid amplification tests (NAATs) to diagnose COVID-19 [1]. NAATs detect the DNA or RNA of pathogens in the sample, and remain the gold standard in the diagnosis of COVID-19 [1]. However, such tests are resource-demanding, requiring trained healthcare providers to collect samples, have a longer assay time (2–8 hours), and are expensive [2, 3]. Therefore, the service capacity of NAATs may not be able to meet the demands when a large number of people need to receive diagnostic testing during the COVID-19 outbreak. Moreover, it takes a long time to receive the testing results, which implies delayed diagnosis and thus delayed prevention and control measures [1].

Rapid antigen tests (RAT) for severe acute respiratory syndrome coronavirus 2 (SARS-CoV-2) have the shortest assay time of 15–20 minutes [2, 4] and are easy to perform as self-testing [4]. With the faster and cheaper diagnosis, RAT could drastically reduce COVID-19 infections and avert potential deaths in several countries [5–7]. Moreover, easy access to RAT has also enabled the reduction of isolation periods at home, thereby saving huge revenue and workdays lost [5, 7].

In response to the fifth wave of the COVID-19 outbreak and the limited service capacity of NAATs, the Hong Kong Department of Health (HKDH) recommends the use of RAT [8]. Use of RAT is voluntary for people with and without COVID-19 symptoms. The guideline stated the procedures to be undertaken when people obtained positive results. First, all people with positive RAT results should report to the Centre for Health Protection within 24 hours via an online platform. Second, they are arranged for a door-to-door delivery of a nasal swab self-sampling kit by courier service for NAATs. Positive cases confirmed by NAATs are then triaged based on risk assessment and appropriate protocol taken and care given [8]. Since older adults are at higher risk of COVID-19, the government also recommend older adults to undergo frequent SARS-CoV-2 testing even if they did not have any symptoms for early detection and timely treatment [9]. Older adults who have received at least two doses of COVID-19 vaccination are recommended to undergo an RAT at least once per week, and those who were not fully vaccinated should do it at least three times a week [9]. Since the fifth wave of the COVID-19 outbreak (31 December 2021), 45% of the approximate 1.6 million cases reported in Hong Kong have been initially detected using RAT [10].

This study focused on older adults. Increasing age is a leading risk factor for severe COVID-19 cases and mortality [11]. The majority of the severe COVID-19 cases and associated deaths occurred in individuals aged 60 years or above during the fifth wave of the COVID-19 outbreak in Hong Kong [10]. However, compared to their younger counterparts, older adults are less likely to adopt health technology innovations [12, 13]. Older adults may experience more difficulties when using such innovations, in this case using RAT [13]. People's acceptance of RAT may influence its intended public health benefits. To our knowledge, limited studies explored the usage or acceptance of RAT and its facilitators and barriers. Rosella et al.'s study indicated high acceptance of high-frequency RAT use at workplaces in Canada due to the strong belief that it contributes to workplace and community safety [14]. Regarding facilitators of RAT use, previous studies have highlighted the desire for certainty and reassurance, incentives, motivation to meet but protect close people or people with a higher risk of severe COVID-19 infections, the possibility to take part in leisure activities and to avoid getting infected and infecting, amongst others as key factors [14, 16]. Nonetheless, several barriers have also been reported, including issues of availability, access and cost, especially in resource-restricted communities, as well as lack of knowledge and misconceptions about RAT,

intolerance of the collection methods (i.e., nasal vs. Nasopharyngeal swab), and doubts regarding the validity of the tests, amongst others [15–19].

In support of RAT use, the Hong Kong government produces some relevant health promotional materials regarding RAT use, but no evaluation has been done to find out whether older adults find them useful. Nonetheless, previous studies found associations between satisfaction with health promotional materials and COVID-19 vaccine uptake [20, 21]. Moreover, misinformation about COVID-19 and its interventions, like testing and vaccination, is widespread, and thoughtful consideration could mitigate the negative impact of misinformation [22]. Thoughtful consideration of COVID-19 information was found to be a facilitator to receiving COVID-19 vaccination among the Chinese population, and the same may apply to RAT use [23].

To our knowledge, no studies looked at RAT use or acceptance among older adults. To address the knowledge gaps, this study investigated the behavioral intention to use RAT and associated factors among Hong Kong older adults aged 65 years or above in two scenarios; i) using RAT regularly for health status monitoring without any symptoms and ii) RAT use when having COVID-19-like symptoms.

## Material and methods

### Ethics statement

The study was conducted according to the guidelines of the Declaration of Helsinki, and was approved by the Survey and Behavioral Research Ethics Committee (reference: SBRE-19-187, date of approval: 10 December 2020). Informed consent was obtained from all subjects involved in the study.

### Study design

This was a cross-sectional random telephone survey conducted among community-dwelling Chinese-speaking individuals aged 65 years or above in Hong Kong between May 11 and 11 July, 2022. During the study period, the daily-confirmed COVID-19 cases slowly increased from 273 on May 11, 2022 to 2,992 on July 11, 2022. The COVID-19 situation in Hong Kong during the study period was presented in **Fig 1**.

### Participants and data collection

Participants were community-dwelling Chinese-speaking individuals aged ≥65 years who had a Hong Kong ID card. Those who were not able to communicate effectively with the study interviewers were excluded. The data collection methods were identical to our published study [20]. First, we input all household telephone numbers listed in the most updated telephone directories (about 350,000) into an Excel file. A total of 4,000 numbers were randomly selected from the file by using the function of "select random cells". Experienced interviewers carried out the telephone interviews 6–10 pm during weekdays and 2–9 pm on Saturdays to avoid under-sampling of working individuals. If no one answered five calls made at different time slots, we would consider such a household to be non-valid (i.e., one without an eligible participant). If there was more than one person aged ≥65 years in the household, the interviewer invited the person whose last birthday was closest to the survey date to participate in the survey. This practice was done to avoid contamination and introducing additional confounding factors. Participants were screened for eligibility, briefed about the study, and guaranteed anonymity. Participants were also assured that participation was voluntary and that opting-out of the study or declining to participate would have no consequences. No sensitive identifying information was collected from the participants.

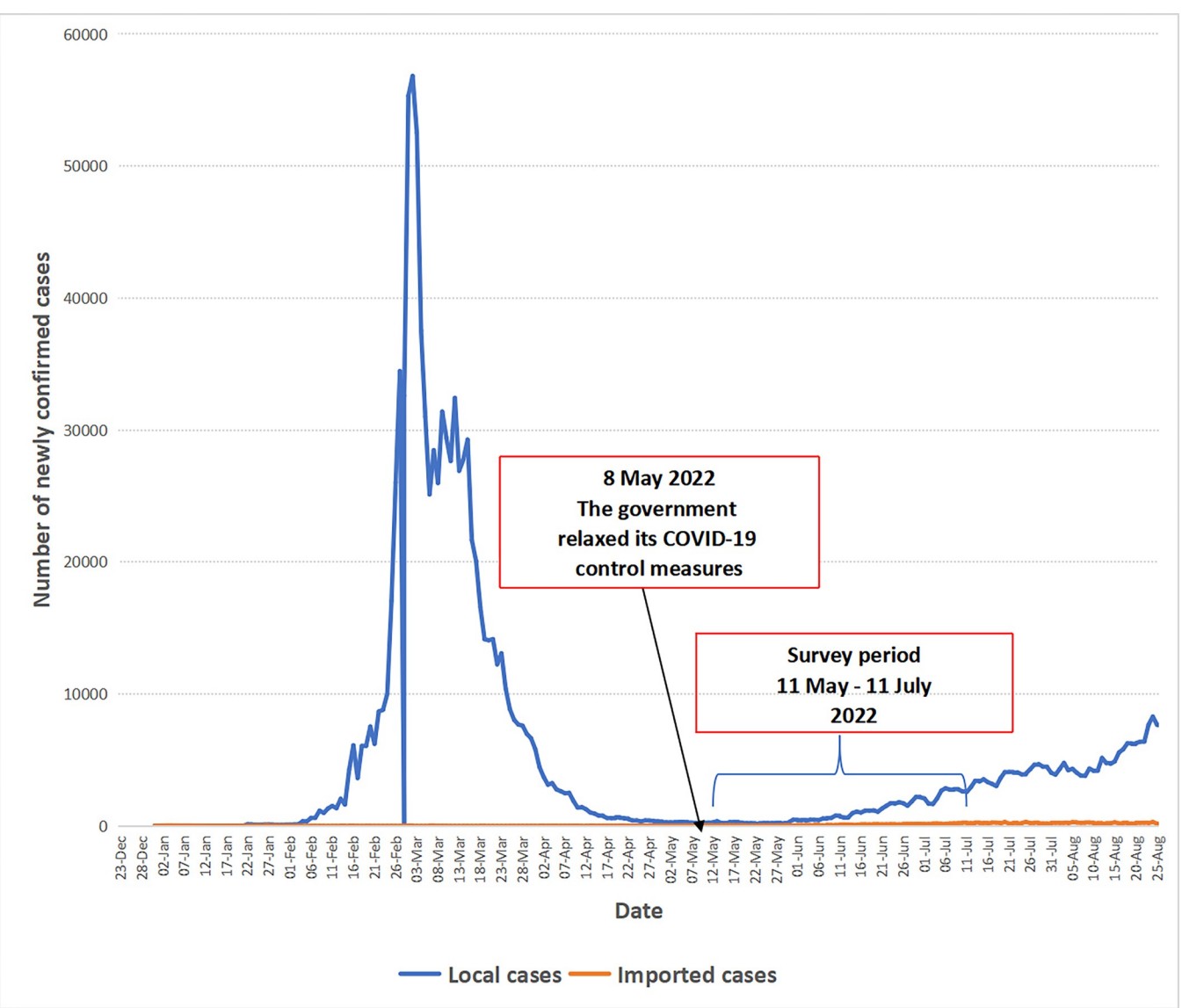

**Fig 1. The COVID-19 situation in Hong Kong during the study period.**

This study was a random telephone survey and there was no face-to-face contact between the interviewers and the participants. Therefore, verbal instead of written informed consent was obtained. The interviewers signed a form pledging the participants had been fully informed about the study. The first part of the form was a checklist to confirm whether the following contents have been introduced. They were: 1) research purpose, 2) research process, 3) main contents of the survey, 4) time required for completing the survey, 5) the rights of the participant, 6) the interviewer guaranteed that the denial to participate would have no consequences, and 7) the interviewer had explained the confidentiality of the research data. The interviewer then documented whether the participants fully understand the above-mentioned contents and whether they verbally expressed willingness to participate in the study in the same form. Questions raised by the participants about the study were also recorded on the form. Finally, the interviewers signed their names with date at the bottom of the form. Such

procedures were approved by the Survey and Behavioral Research Committee of the Chinese University of Hong Kong. Same procedures to obtain verbal informed consent have been used in studies without face-to-face contacts [24–26].

Each telephone interview lasted for approximately 20 minutes. We contacted 3,840 households, of which 625 contained eligible older adults, 255 declined to participate in the survey, and 370 completed the telephone survey, giving a response rate of 59%. No incentives were offered to study participants, and ethical clearance was given by the Survey and Behavioral Research Ethics Committee of The Chinese University of Hong Kong (SBRE-19-187).

## Measurements

### Development of the questionnaire

To better understand older adult's perspectives and experience with RAT, we conducted in-depth interviews with seven community-dwelling Chinese-speaking individuals aged ≥65 years (2 males and 5 females). All informants had prior experience using RAT. We identified three themes related to facilitators: (1) convenience, (2) concern about COVID-19 infection when waiting for the NAT tests, and (3) early identification of COVID-19 infection through regular RAT. Another four themes related to the barriers were identified. They were: (1) concern about the accuracy of RAT, (2) difficulty to select a suitable one from so many brands of RAT on the market, (3) difficulty to use RAT, and (4) did not know what to do if having a positive RAT result. Based on the findings of the interviews, a panel of researchers in public health, behavioral health, and healthy psychology developed the questionnaire. To test the questionnaire's readability and clarity, we piloted it with 10 senior citizens. They all agreed that the questionnaire's length and readability were appropriate. These older adults were not part of the final survey. The questionnaire was then revised and completed by the panel.

### Background characteristics

Participants provided information about their sociodemographics, presence of chronic disease, history of COVID-19, and received doses of the COVID-19 vaccine.

### Behaviors and behavioral intention to use RAT

Participants were asked whether they had used RAT. We further asked for some details among those with experience of using RAT. We also asked about their likelihood of using RAT regularly (i.e., every one or few weeks) when they do not have any symptoms and the likelihood of using RAT when they had some COVID-19-like symptoms such as fever and/or cough. The response categories ranged from 1 = very unlikely, 2 = unlikely, 3 = neutral, 4 = likely, to 5 = very likely. We dichotomized the responses and defined behavioral intention as "likely" or "very likely". The same definition of behavioral intention was commonly used in published studies [23, 27].

### Perceptions related to RAT

The Theory of Planned Behavior (TPB) was used as the framework to measure perceptions related to RAT [28]. Based on the findings of the in-depth interviews, three items were constructed in this study to measure positive attitudes toward RAT (e.g., RAT is convenient for you). Another four items were constructed to measure negative attitudes toward RAT (e.g., RAT is expensive for you). In addition, two items measured perceived subjective norm, and another two items measured perceived behavioral control related to RAT. The Positive Attitude Scale, the Negative Attitude Scale, the Perceived Subjective Norm Scale, and the Perceived

Behavioral Control Scale were constructed by summing up individual item scores (1 = disagree, 2 = neutral, & 3 = agree). The Cronbach's alpha of these scales ranged from 0.61 to 0.81. Single factors were identified by exploratory factor analysis, explaining 64.5–71.9% of the total variance.

### Satisfaction of RAT health promotional materials (e.g., advertisements, posters, and others) produced by the government

We adapted validated items measuring satisfaction with COVID-19 vaccination health promotional materials produced by the government among older adults in Hong Kong [21]. We replaced the phrase "COVID-19 vaccination" with "RAT". The responses to the items were 1 = no, 2 = uncertain, and 3 = yes.

### Thoughtful consideration of the veracity of COVID-19-specific information

A single item validated in the Chinese population was used to measure the frequency of thoughtful consideration of the veracity of COVID-19-specific information obtained from different channels (e.g., TV, radio, newspaper, Internet) in the past month (response categories: 1 = almost none, 2 = seldom, 3 = sometimes, & 4 = always) [23, 29].

### Sample size planning

The target sample size was 360. We assumed that 50% of the older adults intended to use RAT under different conditions. Assuming a 10–40% prevalence of behavioral intention in the reference group (with no facilitating condition), the target sample size could identify a minimum odds ratio of 1.76 between those with and without a facilitating condition (Power: 0.80, alpha value: 0.05; PASS 11.0, NCSS LLC). In 2021, 18.2% (1.36 million) of Hong Kong residents were ≥65 years [30]. Assuming a response rate of 55–60% of eligible households, field staff would need to screen approx. 3,600 households to reach the target sample size.

### Statistical analysis

Descriptive statistics of all variables were presented. We calculated Cronbach's alphas using reliability tests and used principal component analysis with varimax rotation to perform exploratory factor analysis. Behavioral intention to use RAT regularly to monitor one's health status when they did not have any symptoms and when they had COVID-19-like symptoms were dependent variables. We first fitted univariate logistic regression models to assess the significance between background characteristics and the dependent variables. We then fitted a single logistic regression model involving all significant background characteristics and one independent variable of interest at a time. We obtained the crude odds ratios (ORs), adjusted odds ratios (AORs), and their 95% confidence intervals (CIs). Data analysis was performed using SPSS 26.0 (IBM Corp., Armonk, NY, USA) and statistical significance was considered at $p < 0.05$.

## Results

### Background characteristics of the participants

The majority of participants were below 75 years of age (83%), female (60.8%), married or cohabited with a partner (74.6%), without tertiary education (89.2%) or full-time or part-time work (86.2%), and had a monthly household income below HK$20,000 (USD 2580) (74.2%) (Table 1).

**Table 1. Background characteristics of the participants (n = 370).**

| Characteristics | n | % |
|---|---|---|
| **Sociodemographic characteristics** | | |
| Age group, years | | |
| 65–69 | 182 | 49.2 |
| 70–74 | 125 | 33.8 |
| 75 or above | 63 | 17.0 |
| Gender | | |
| Male | 145 | 39.2 |
| Female | 225 | 60.8 |
| Relationship status | | |
| Currently single | 94 | 25.4 |
| Married or cohabited with a partner | 276 | 74.6 |
| Education level | | |
| Primary or below | 157 | 42.4 |
| Secondary | 173 | 46.8 |
| Tertiary or above | 40 | 10.8 |
| Current employment status | | |
| Unemployed/retired/housewife | 319 | 86.2 |
| Full-time/part-time | 51 | 13.8 |
| Monthly household income, HK$ (US$) | | |
| <20,000 (2580) | 273 | 74.2 |
| ≥20,000 (2580) | 49 | 13.3 |
| Refuse to disclose | 46 | 12.5 |
| Receiving Comprehensive Social Security Assistance (CSSA) | | |
| No | 342 | 92.4 |
| Yes | 28 | 7.6 |
| Living alone | | |
| No | 304 | 82.2 |
| Yes | 66 | 17.8 |
| **Presence of chronic conditions, yes** | | |
| Hypertension | 173 | 46.8 |
| Chronic cardiovascular diseases | 40 | 10.8 |
| Chronic lung diseases | 6 | 1.6 |
| Chronic liver diseases | 8 | 2.2 |
| Chronic kidney diseases | 2 | 0.5 |
| Diabetes Mellitus | 70 | 18.9 |
| Any of above | 223 | 60.3 |
| **History of COVID-19 and COVID-19 vaccination** | | |
| History of COVID-19 | | |
| No | 276 | 74.6 |
| Yes | 94 | 25.4 |
| Number of doses of COVID-19 vaccination received by the participants | | |
| 0–1 | 29 | 7.8 |
| 2 | 123 | 33.2 |
| 3–4 | 218 | 58.9 |

### Behaviors and behavioral intention to use RAT

Among the respondents, 90.3% had ever used RAT. The common reasons for RAT use were monitoring health status (51.5%), compulsory testing enforced by the government (31.7%), and having COVID-19-like symptoms (23.4%). Among RAT users (n = 334), 21.6% obtained positive results. The common difficulties faced when using RAT included difficulty to choose a RAT kit (49.4%), uncertainty of how to use RAT (3.3%) and not knowing what to do after getting a positive result (3.6%).

Among all participants, 27.3% intended to use RAT regularly for health status monitoring when they did not have any symptoms, while 87.0% intended to use RAT when they had COVID-19-like symptoms (**Table 2**). Additional analyses was done to compare the level of intention before and after June 30, 2022, when the government stopped providing free RAT kits. The results showed that behavioral intention to use RAT kits with some symptoms was significantly lower after June 2022, as compared to the time before June 2022 (60% versus 88.2%, p < .001). However, the level of behavioral intention to use RAT kits without any symptoms was not statistically different before and after June 2022 (27.9% versus 13.3%, p = .22).

### Factors associated with behavioral intention to use RAT regularly for health status monitoring when they do not have any symptoms

In univariate analysis, having a full-time/part-time job was associated with a higher intention to use RAT regularly for health status monitoring when they do not have any symptoms (**Table 3**). After adjustment for significant background characteristics, those with more positive attitudes toward RAT (AOR: 2.16, 95%CI: 1.32, 3.54), and perceived Hong Kong government and their children and/or other family members would support them using RAT (AOR: 1.66, 95%CI: 1.13, 2.43) had more behavioral intention. A negative association was found between negative attitudes toward RAT and the dependent variable (AOR: 0.86, 95%CI: 0.75, 0.98). In addition, the belief that the RAT health promotion materials were helpful for them to understand how to use RAT (AOR: 2.22, 95%CI:1.03, 4.79), and thoughtful consideration of the veracity of COVID-19-specific information obtained from different channels (AOR:1.27, 95%CI: 1.02, 1.58) were also positively associated with this dependent variable (**Table 4**).

### Factors associated with behavioral intention to use RAT when they had COVID-19-like symptoms

In univariate analysis, educational level and history of using RAT age were associated with higher intention to use RAT in case of COVID-19-like symptoms, while age was associated with less intention (**Table 3**). In the adjusted model, those with more positive attitudes toward RAT (AOR: 1.84, 95%CI: 1.35, 2.52), and perceived Hong Kong government and their children and/or other family members would support them using RAT (AOR: 2.41, 95%CI: 1.69, 3.43) were associated with higher odds of behavioral intention. In addition, those with thoughtful consideration of the veracity of COVID-19-specific information obtained from different channels (AOR: 1.70, 95%CI: 1.24, 2.33) also had a positive association (**Table 4**).

## Discussion

### Principal findings

This is one of the first studies exploring the facilitators and barriers to using RAT among older adults. It was based on a random and population-based sample and expanded the application of the Theory of Planned Behaviour (TPB), which were strengths of this study. The study revealed that RAT was widely used by older adults (>90%) during the fifth wave of the

**Table 2. Behaviors and perceptions related to rapid antigen test for COVID-19 (RAT) (n = 370).**

| Characteristics | n | % |
|---|---|---|
| **Behaviors related to RAT** | | |
| History of using RAT | | |
| No | 36 | 9.7 |
| Yes | 334 | 90.3 |
| Reasons for using RAT (among 334 participants with experiences of using RAT) | | |
| Compulsory testing enforced by the government | 106 | 31.7 |
| You and/or your family members having flu-like symptoms | 78 | 23.4 |
| Someone around you infected with COVID-19 | 50 | 15.0 |
| Monitor your health status regularly | 172 | 51.5 |
| Other reasons | 117 | 35.0 |
| Obtained positive results by using RAT (among 334 participants with experiences of using RAT) | | |
| No | 262 | 78.4 |
| Yes | 72 | 21.6 |
| Difficulties encountered when using RAT | | |
| Difficulty to choose a RAT kit | 165 | 49.4 |
| Not sure about how to use RAT | 11 | 3.3 |
| Difficulty to interpret RAT results | 7 | 2.1 |
| Not sure how to do after having a positive RAT result | 12 | 3.6 |
| Not sure how to properly dispose of the used RAT kits | 6 | 1.8 |
| Others | 3 | 0.9 |
| No difficulties | 159 | 47.6 |
| **Behavioral intention to use RAT** | | |
| Likelihood of using RAT regularly (every one or few weeks) to monitor your health status when you do not have any symptoms | | |
| Very unlikely/unlikely/neutral | 269 | 72.7 |
| Likely/very likely | 101 | 27.3 |
| Likelihood of using RAT when you have some flu-like symptoms (e.g., fever, cough) | | |
| Very unlikely/unlikely/neutral | 48 | 13.0 |
| Likely/very likely | 322 | 87.0 |
| **Perceptions related to RAT** | | |
| Positive attitudes toward RAT, agree | | |
| RAT is convenient for you | 331 | 89.5 |
| Using RAT can reduce the risk of COVID-19 infection caused by queuing for the COVID-19 nuclear acid testing | 319 | 86.2 |
| Using RAT regularly can identify COVID-19 infection earlier and reduce the risk of transmitting COVID-19 to others | 331 | 89.5 |
| Positive Attitude Scale [1], mean (SD) | 8.6 | 0.9 |
| Negative attitudes toward RAT, agree | | |
| RAT is expensive for you | 50 | 13.5 |
| You do not know how to choose a reliable RAT kit | 232 | 62.7 |
| You concern about the accuracy of RAT | 69 | 18.6 |
| You do not know what to do if receiving a positive RAT result | 53 | 14.3 |
| Negative Attitude Scale [2], mean (SD) | 7.0 | 1.8 |
| Perceived subjective norm related to RAT, agree | | |
| Hong Kong government would support you to use RAT regularly | 317 | 85.7 |
| Your children and/or other family members would support you to use RAT | 274 | 74.1 |
| Perceived Subjective Norm Scale [3], mean (SD) | 5.5 | 0.8 |
| Perceived behavioral control to use RAT, agree | | |

(*Continued*)

**Table 2.** (Continued)

| Characteristics | n | % |
|---|---|---|
| You are confident to use RAT kit properly | 325 | 87.8 |
| You are confident to receive governmental support services for people with COVID-19 after having a positive RAT result | 232 | 62.7 |
| Perceived Behavioral Control Scale [4], mean (SD) | 5.3 | 1.0 |
| **Satisfaction with RAT health promotional materials (e.g., advertisements, posters, and others) produced by the government** | | |
| Whether the information is easy to understand | | |
| No/uncertain | 134 | 36.2 |
| Yes | 236 | 63.8 |
| Whether the materials are helpful for you to understand how to use RAT | | |
| No/uncertain | 64 | 17.3 |
| Yes | 306 | 82.7 |
| Whether the materials are helpful for you to understand the procedures to report RAT positive results and to obtain services for people with COVID-19 | | |
| No/uncertain | 116 | 31.4 |
| Yes | 254 | 68.6 |
| **Thoughtful consideration of the veracity of COVID-19-specific information** | | |
| Frequency of thoughtful consideration of the veracity of COVID-19-specific information obtained from different channels (e.g., TV, radio, newspaper, Internet) in the past month | | |
| Almost none | 92 | 24.9 |
| Seldom | 80 | 21.6 |
| Sometimes | 104 | 28.1 |
| Always | 94 | 25.4 |
| Item score, mean (SD) | 2.5 | 1.1 |

[1] Positive Attitude Scale, 3 items, Cronbach's alpha: 0.79, one factor was identified by exploratory factor analysis, explaining for 70.4% of total variance

[2] Negative Attitude Scale, 4 items, Cronbach's alpha: 0.81, one factor was identified by exploratory factor analysis, explaining for 66.5% of total variance

[3] Perceived Subjective Norm Scale, 2 items, Cronbach's alpha: 0.61, one factor was identified by exploratory factor analysis, explaining for 71.9% of total variance

[4] Perceived Behavioral Control Scale, 2 items, Cronbach's alpha: 0.73, one factor was identified by exploratory factor analysis, explaining for 64.5% of total variance

COVID-19 outbreak in Hong Kong. Most of the participants used RAT when having COVID-19-like symptoms, after having contacts with confirmed cases, or for regular health screening as recommended by the Hong Kong government [8, 10, 31]. From April to the end of June 2022, free RAT kits were supplied by the government to support and encourage the elderly to conduct voluntary self-testing regularly for early identification of infected persons and curbing the community transmission chains. Our findings suggested that such an approach had successfully achieved high RAT coverage among older adults, as the level of behavioral intention to use RAT was significant higher when the government was supplying free test kits. Moreover, 30% of the participants used RAT to complete compulsory COVID-19 testing enforced by the government, which might alleviate the burden of NAATs services during the outbreak. Notably, around 20% of the RAT users obtained positive results, implying RAT did play a significant role in the early identification of COVID-19 infection among local older adults.

Difficulty in selecting a reliable RAT kit from the many brands available on the market was the main difficulty encountered by local older adults when using RAT. In early 2022, a number

**Table 3. Associations between background characteristics and behavioral intention to use RAT under different conditions (n = 370).**

| | Intention to use RAT regularly to monitor your health status when you do not have any symptoms | | Intention to use RAT when you have some COVID-19-like symptoms | |
|---|---|---|---|---|
| | OR (95%CI) | P values | OR (95%CI) | P values |
| Age group, years | | | | |
| 65–69 | 1.0 | | 1.0 | |
| 70–74 | 0.69 (0.44, 1.16) | .16 | 1.06 (0.51, 2.23) | .87 |
| 75 or above | 0.63 (0.32, 1.23) | .17 | 0.40 (0.19, 0.83) | .01 |
| Gender | | | | |
| Male | 1.0 | | 1.0 | |
| Female | 0.82 (0.52, 1.31) | .41 | 1.83 (0.99, 3.37) | .052 |
| Relationship status | | | | |
| Currently single | 1.0 | | 1.0 | |
| Married or cohabited with a partner | 0.65 (0.39, 1.07) | .09 | 0.64 (0.30, 1.38) | .26 |
| Education level | | | | |
| Primary or below | 1.0 | | 1.0 | |
| Secondary | 0.98 (0.61, 1.59) | .95 | 2.67 (1.33, 5.37) | .01 |
| Tertiary or above | 0.53 (0.22, 1.28) | .16 | 1.02 (0.41, 2.55) | .96 |
| Current employment status | | | | |
| Unemployed/retired/housewife | 1.0 | | 1.0 | |
| Full-time/part-time | 2.31 (1.25, 4.24) | .01 | 0.77 (0.34, 1.76) | .54 |
| Monthly household income, HK$ (US$) | | | | |
| <20,000 (2580) | 1.0 | | 1.0 | |
| ≥20,000 (2580) | 1.54 (0.81, 2.95) | .19 | 1.16 (0.46, 2.91) | .75 |
| Refuse to disclose | 1.14 (0.57, 2.29) | .71 | 1.70 (0.58, 5.01) | .34 |
| Receiving Comprehensive Social Security Assistance (CSSA) | | | | |
| No | 1.0 | | 1.0 | |
| Yes | 1.53 (0.68, 3.44) | .30 | 0.89 (0.29, 2.67) | .83 |
| Living alone | | | | |
| No | 1.0 | | 1.0 | |
| Yes | 1.00 (0.55, 1.82) | .99 | 1.10 (0.49, 2.47) | .82 |
| Presence of any chronic conditions | | | | |
| No | 1.0 | | 1.0 | |
| Yes | 1.35 (0.84, 2.17) | .22 | 0.90 (0.48, 1.68) | .74 |
| History of COVID-19 | | | | |
| No | 1.0 | | 1.0 | |
| Yes | 0.98 (0.57, 1.70) | .96 | 1.15 (0.55, 2.43) | .71 |
| Number of doses of COVID-19 vaccination received by the participants | | | | |
| 0–1 | 1.0 | | 1.0 | |
| 2 | 1.29 (0.45, 3.70) | .64 | 1.22 (0.41, 3.60) | .73 |
| 3–4 | 2.27 (0.83, 6.20) | .11 | 1.61 (0.56, 4.60) | .38 |
| History of using RAT | | | | |
| No | 1.0 | | 1.0 | |
| Yes | N.A. | < .001 | 7.55 (3.56, 16.01) | < .001 |

OR: crude odds ratios

CI: confidence interval

**Table 4. Factors associated with behavioral intention to use RAT under different conditions (n = 370).**

| | Intention to use RAT regularly to monitor your health status when you do not have any symptoms | | Intention to use RAT when you have some COVID-19-like symptoms | |
|---|---|---|---|---|
| | AOR (95% CI) | P values | AOR (95% CI) | P values |
| **Perceptions related to RAT** | | | | |
| Positive Attitude Scale | 2.16 (1.32, 3.54) | .002 | 1.84 (1.35, 2.52) | < .001 |
| Negative Attitude Scale | 0.86 (0.75, 0.98) | .03 | 0.90 (0.75, 1.08) | .24 |
| Perceived Subjective Norm Scale | 1.66 (1.13, 2.43) | .01 | 2.41 (1.69, 3.43) | < .001 |
| Perceived Behavioral Control Scale | 1.14 (0.86, 1.51) | .35 | 1.29 (0.93, 1.78) | .13 |
| **Satisfaction with RAT health promotional materials (e.g., advertisements, posters, and others) produced by the government** | | | | |
| Whether the information is easy to understand | | | | |
| No/uncertain | 1.0 | | 1.0 | |
| Yes | 1.59 (0.95, 2.66) | .08 | 1.31 (0.67, 2.57) | .43 |
| Whether the materials are helpful for you to understand how to use RAT | | | | |
| No/uncertain | 1.0 | | 1.0 | |
| Yes | 2.22 (1.03, 4.79) | .04 | 1.71 (0.79, 3.67) | .17 |
| Whether the materials are helpful for you to understand the procedures to report RAT positive results and to obtain services for people with COVID-19 | | | | |
| No/uncertain | 1.0 | | 1.0 | |
| Yes | 1.46 (0.85, 2.52) | .17 | 1.79 (0.91, 3.53) | .09 |
| Frequency of thoughtful consideration of the veracity of COVID-19-specific information obtained from different channels (e.g., TV, radio, newspaper, Internet) in the past month | | | | |
| Item score | 1.27 (1.02, 1.58) | .02 | 1.70 (1.24, 2.33) | < .001 |

AOR: adjusted odds ratios, odds ratios adjusted for significant background characteristics listed in Table 3

of RAT kits became available in Hong Kong. Some older adults may experience "choice overload" and are uncertain about which option to take when there are many competing options [32]. Similarly, choice overload was also a barrier for local older adults to receive COVID-19 vaccination [21]. In response to this situation, the Hong Kong government quickly released a list of accepted RAT kits. In contrast to our hypothesis, relatively few older adults encountered difficulties with follow-up procedures after receiving a positive result. The government spent lots of effort explaining such procedures through multiple mass media channels and established hotlines supporting self-testing users [8, 31]. Nonetheless, additional targeted efforts in community education are needed to cater to older adults who may be unsure about the next course of action after receiving a positive RAT result.

It is encouraging that most older adults intended to use RAT when they have COVID-19-like symptoms, which corroborates with the WHO recommendations [1, 17]. However, very few intended to use RAT for regular health monitoring when they did not have any symptoms. One possible explanation is that the COVID-19 pandemic was stable during the study

period. Older adults may perceive a low risk of COVID-19 and hence believe that it is not necessary to perform regular screening. Moreover, the government's free supply of RAT kits for older adults stopped by the end of June 2022. Older adults had to buy the kits themselves. The cost of RAT might reduce their motivation to use it for regular screening.

Participants with a history of using RAT were more likely to use RAT under both conditions. One possible explanation is that previous users are more familiar with RAT, and such finding is similar to self-testing for other diseases like HIV [33, 34]. This implies that future programs promoting RAT should give more attention to older adults who are first-time users of RAT. Having a full-time or part-time job was positively associated with RAT use for health monitoring among our participants. This is because full-time and part-time employees were more likely to be subjected to compulsory regular testing for workplace safety precautions. Moreover, when having COVID-19-like symptoms, barriers and lower usage were more among those with older age and lower education level. This is probably because such demographics are less likely to adopt health innovations such as RAT use [12, 13]. This, thus, implies that the RAT instruction details on product leaflets should be revised and made easily understood by older adults with low literacy levels.

## Implications of study findings

The TPB is a useful framework to understand factors associated with behavioral intention to use RAT, as three of its four constructs (i.e., positive attitudes, negative attitudes, and perceived subjective norm) were significantly associated with the dependent variables. Our findings also provided some practical implications to inform future RAT implementation. First, the majority of the participants had a positive attitude toward RAT. Strengthening such positive attitudes would, hence, be useful to increase RAT usage among older adults, as it was associated with a higher likelihood of using RAT under both scenarios. Health communication messages should, in addition, emphasize RAT as a convenient alternative option to NAATs for early case identification for older adults. Second, involving children and family members of older adults in health promotion might be another useful strategy, as their support was important for older adults to use RAT. The government should also provide continuous support for RAT among older adults. Moreover, concerns about the cost and accuracy of RAT kits, difficulties in choosing a reliable kit and not knowing the follow-up procedures after having positive results are barriers to the use of RAT for health monitoring among our participants. The government should consider supplying free or subsidized RAT kits for older adults, especially those with low-income levels, to increase accessibility. Updated information about the accuracy and a list of acceptable kits should be disseminated to the public. There is also a need to improve RAT health promotional materials, as some older adults found them unhelpful and difficult to understand. Therefore, providing step-to-step instructions and demonstration videos would help enable older adults to understand how to use RAT. Our findings also highlighted the role of thoughtful consideration of the veracity of information specific to COVID-19 and RAT. Thoughtful consideration would mitigate the negative impacts of misinformation on RAT use. In corroboration, previous studies showed that thoughtful consideration of information was associated with lower COVID-19 vaccine hesitancy in different populations [23].

## Limitations

This study had some limitations. First, some measurement tools (e.g., attitudes toward RAT) were constructed for our study, as there were no validated tools for older adults in Hong Kong. However, the reliability of these measurements was acceptable both in the pilot and actual

study. Second, we also did not include older residents of residential care homes, and different determinants might apply to this group, making our findings not generalizable to the entire population of older adults in Hong Kong. Third, selection bias existed due to non-response, as we were not able to collect information from those who refused to participate in the study. Additionally, although the Hong Kong's residential landline penetration rate was 73.26% and landline ownership was higher among elderly than that of the younger generation [35], the database used to select study participants could not cover all older residents in Hong Kong. Moreover, recall bias also existed as data were self-reported and verification was not feasible, and causality could not be established due to the cross-sectional design of this study. Nonetheless, the study provides useful information on facilitators and barriers to RAT use among older adults, despite the limitations.

## Conclusions

In this study, the majority of the community-dwelling older adults aged ≥65 years had used RAT before and a good number obtained positive COVID-19 results. Future health promotion should facilitate older adults to select a reliable RAT kit. Strengthening positive attitudes and involving recommendation from Hong Kong government and support from their children and/or other family members might be crucial considerations to increase the use of RAT in this population group. Moreover, more efforts to empower older adults with adequate skills to evaluate the veracity of information about RAT and COVID-19, in general, are needed.

## Acknowledgments

The authors would like to express their gratitude to all subjects for their engagement in this study.

## Author Contributions

**Conceptualization:** Joseph Kawuki, Yuan Fang, Zixin Wang.

**Data curation:** Yuan Fang, Fuk-yuen Yu, Danhua Ye, Zixin Wang.

**Formal analysis:** Joseph Kawuki, Zixin Wang.

**Methodology:** Yuan Fang, Fuk-yuen Yu, Danhua Ye, Zixin Wang.

**Project administration:** Yuan Fang, Fuk-yuen Yu, Danhua Ye.

**Supervision:** Zixin Wang.

**Writing – original draft:** Joseph Kawuki, Paul Shing-fong Chan, Siyu Chen.

**Writing – review & editing:** Yuan Fang, Fuk-yuen Yu, Paul Shing-fong Chan, Siyu Chen, Zixin Wang.

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
