## [Decision Letter · Decision Letter 0]

19 Jun 2023

PGPH-D-23-00780

Facilitators and barriers to use Rapid Antigen Test among community-dwelling older adults in Hong Kong: A Population-Based Random Telephone Survey

Dear Dr. Wang,

Thank you for submitting your manuscript to PLOS Global Public Health. After careful consideration, we feel that it has merit but does not fully meet PLOS Global Public Health’s publication criteria as it currently stands. Therefore, we invite you to submit a revised version of the manuscript that addresses the points raised during the review process.

We look forward to receiving your revised manuscript.

Kind regards,

Sanghyuk S Shin

Academic Editor

Journal Requirements:

1. In the ethics statement in the Methods, you have specified that verbal consent was obtained. Please provide additional details regarding how this consent was documented and witnessed, and state whether this was approved by the IRB.

- https://doi.org/10.3390/vaccines10060966

In your revision ensure you cite all your sources (including your own works), and quote or rephrase any duplicated text outside the methods section. Further consideration is dependent on these concerns being addressed.

3. Please amend your online detailed Financial Disclosure statement. This is published with the article. It must therefore be completed in full sentences and contain the exact wording you wish to be published.

a) State the initials, alongside each funding source, of each author to receive each grant. For example: "This work was supported by the National Institutes of Health (####### to AM; ###### to CJ) and the National Science Foundation (###### to AM)."

4. Please ensure that the funders and grant numbers match between the Financial Disclosure field and the Funding Information tab in your submission form. Note that the funders must be provided in the same order in both places as well.

5. Please update your online Competing Interests statement. If you have no competing interests to declare, please state: “The authors have declared that no competing interests exist.”

6. Please provide separate figure files in .tif or .eps format only and remove any figures embedded in your manuscript file. Please also ensure that all files are under our size limit of 10MB.

Additional Editor Comments (if provided):

Please respond to the comments made by the reviewers. In particular, please note the following:

- I agree with Reviewer #1 that the term "significant other" is not a very specific term. Please replace this with a more specific term.

- I also agree that the title should be changed to specify COVID-19 or SARS-CoV-2.

- One of the recommendations in STROBE is to describe clearly the study design (cohort, cross-sectional, etc.) in the title and/or abstract. Please specify that the study employed a cross-sectional study design in the abstract and Methods section of the main text.

- I also share the same concern as Reviewer #2 regarding potential for selection bias in the current era of cell phone and landline use. Please clarify in Methods whether the database used for sampling contains both cell phone and landline use. Please also provide any information that supports the completeness of this database in covering all residents >=65 years age in Hong Kong. Please address this issue in Limitations, if completeness can not be verified.

Reviewers' comments:

Reviewer's Responses to Questions

**Comments to the Author**

1. Does this manuscript meet PLOS Global Public Health’s publication criteria? Is the manuscript technically sound, and do the data support the conclusions? The manuscript must describe methodologically and ethically rigorous research with conclusions that are appropriately drawn based on the data presented.

Reviewer #1: Yes

Reviewer #2: Yes

2. Has the statistical analysis been performed appropriately and rigorously?

Reviewer #1: Yes

Reviewer #2: Yes

3. Have the authors made all data underlying the findings in their manuscript fully available (please refer to the Data Availability Statement at the start of the manuscript PDF file)?

Reviewer #1: No

Reviewer #2: Yes

4. Is the manuscript presented in an intelligible fashion and written in standard English?

Reviewer #1: Yes

Reviewer #2: Yes

5. Review Comments to the Author

Reviewer #1: Thank you for the opportunity to review this manuscript. The manuscript is well written.

Below a few minor comments

Title: Please add that this is testing for SARS-CoV-2

Results:

Factors

….perceived significant others would support them using RAT (AOR: 1.66, 95%CI: 1.13, 2.43) had more behavioral intention…: can you rephrase this sentence. The term 'perceived significant others' is not clear - you mean individuals who had a life partner?

Discussion:

'Hard to choose a reliable RAT kit is the main difficulty encountered by local older adults when using RAT' - you mean difficulty to choose? Please clarify the statement

It is possible that the availability of free test kits increased someone's intention to test. Did the authors include this into their analyses? Is there any difference between before/after June 2022, when the free test kit program stopped?

Can you clarify what were the guideline from Hong Kong Dept of Health on testing with and without symptoms? Was it mandatory to test without symptoms? - Why would anyone want to test without any symptoms?

Reviewer #2: The study examined the behavioral intentions of older adults in Hong Kong to use rapid antigen tests (RAT) to monitor and test for COVID-19. The authors provide useful information about the acceptability of RAT among a high-risk population group, and offer recommendations to effectively support older adults, particularly during global health emergencies such as the COVID-19 pandemic. Overall, the paper is a good contribution to the literature and the development of public health programs. Below are a few recommendations for improvement.

Sample size planning: 18.2% of Hong Kong residents are aged 65 years and older. What is the total population?

Statistical analysis: In this statement "We then fitted a single logistic regression model involving all significant background characteristics and one independent variable of interest", please clarify "one independent variable of interest". Is this referring to the factors identified from the principal component analysis/exploratory factor analysis?

Results

i) In the first sentence on page 8, elaborate on what "compulsory testing" means. Does this refer to testing enforced by the government/public health officials?

ii) Among the 21.6% of RAT users who tested positive, it would also be interesting to see the percent breakdown of individuals who intended to use RAT regularly for health monitoring (without symptoms) vs those who intended to use RAT when symptomatic.

Discussion

Given that TPB was the guiding framework of the study, it would be good to include a statement or two about the theory and tie that to the individual constructs (i.e., constructs that were found to be significantly associated with the outcome variables) prior to providing recommendations about how these results might inform future health interventions. The authors do partly highlight this in the "Implications" section, however, it would be good to connect to the TPB and also provide a few citations from the literature.

Limitations: These are a couple of points to consider- adding to the issue of selection bias, might people's preference for cell phones over landlines be a potential study limitation, or are landlines still the preferred mode of communication among older adults in Hong Kong? Also, is it possible that some residents may be registered in the national "Do not call" list to avoid unsolicited calls, e.g., telemarketing?

Additional comments: The writing can be improved for clarity. Below are a few suggestions that can be tweaked.

i) On page 5, the sentence "Interviewers screened the eligibility of prospective participants, briefed them about the study, and made guarantees of anonymity, their right to quit at anytime and refusal would have no consequences" can be written as "Participants were screened for eligibility, briefed about the study, and guaranteed anonymity. Participants were also assured that participation was voluntary and that opting-out of the study or declining to participate would have no consequences."

"All informants had the experience of using RAT before." can be written as "All informants had prior experience using RAT."

ii) In the second paragraph on page 9, the first sentence, "Hard to choose a reliable RAT kit is the main difficulty encountered by local older adults when using RAT", can be rewritten as "Difficulty selecting a reliable RAT kit is the main challenge faced by local older adults".

Also, the last sentence in this paragraph can be rephrased from "Nonetheless, more targeted efforts in community education are still needed to eliminate the existing small proportion of older adults facing such difficulties of the uncertainty of what to do in case of positive RAT results" to "Nonetheless, additional targeted efforts in community education are needed to cater to older adults who may be unsure about the next course of action after receiving a positive RAT result."

iii) In the first paragraph on page 10, "never-users" can be replaced by "first-time users of RAT" or "individuals without prior experience using RAT"

iv) Please doublecheck that in-text citations are in the right format and proper numbering, e.g., "Wang et al., 2022b" on page 6 is inconsistent with others.

6. PLOS authors have the option to publish the peer review history of their article (what does this mean?). If published, this will include your full peer review and any attached files.

**Do you want your identity to be public for this peer review?** For information about this choice, including consent withdrawal, please see our Privacy Policy.

Reviewer #1: **Yes: **Chrysovalantis Stafylis

Reviewer #2: No

---

## [Editor Report · Decision Letter 1]

17 Jul 2023

Facilitators and Barriers to use Rapid Antigen Test for SARS-CoV-2 among community-dwelling older adults in Hong Kong: A Population-based Cross-sectional Random Telephone Survey

PGPH-D-23-00780R1

Dear Professor Wang,

We are pleased to inform you that your manuscript 'Facilitators and Barriers to use Rapid Antigen Test for SARS-CoV-2 among community-dwelling older adults in Hong Kong: A Population-based Cross-sectional Random Telephone Survey' has been provisionally accepted for publication in PLOS Global Public Health.

Best regards,

Sanghyuk S Shin

Academic Editor